# A Review of Microrobot’s System: Towards System Integration for Autonomous Actuation In Vivo

**DOI:** 10.3390/mi12101249

**Published:** 2021-10-15

**Authors:** Zhongyi Li, Chunyang Li, Lixin Dong, Jing Zhao

**Affiliations:** 1School of Mechatronical Engineering, Beijing Institute of Technology, Beijing 100081, China; zhongyili@bit.edu.cn (Z.L.); cyli@bit.edu.cn (C.L.); 2Beijing Advanced Innovation Center for Intelligent Robots and Systems, Beijing Institute of Technology, Beijing 100081, China; 3Department of Biomedical Engineering, City University of Hong Kong, Kowloon Tong, Hong Kong 999077, China; lixidong@cityu.edu.hk

**Keywords:** microrobot, autonomous actuation system, medical imaging, system integration, in vivo

## Abstract

Microrobots have received great attention due to their great potential in the biomedical field, and there has been extraordinary progress on them in many respects, making it possible to use them in vivo clinically. However, the most important question is how to get microrobots to a given position accurately. Therefore, autonomous actuation technology based on medical imaging has become the solution receiving the most attention considering its low precision and efficiency of manual control. This paper investigates key components of microrobot’s autonomous actuation systems, including actuation systems, medical imaging systems, and control systems, hoping to help realize system integration of them. The hardware integration has two situations according to sharing the transmitting equipment or not, with the consideration of interference, efficiency, microrobot’s material and structure. Furthermore, system integration of hybrid actuation and multimodal imaging can improve the navigation effect of the microrobot. The software integration needs to consider the characteristics and deficiencies of the existing actuation algorithms, imaging algorithms, and the complex 3D working environment in vivo. Additionally, considering the moving distance in the human body, the autonomous actuation system combined with rapid delivery methods can deliver microrobots to specify position rapidly and precisely.

## 1. Introduction

Microrobots are robots on a microscale that can perform high-precision operations in micro spaces. Considering the limitations of existing invasive medical devices and traditional surgical robots, it is hoped that microrobots will enter the complex and narrow areas of the human body, especially the far ends of cerebrovascular vessels and bile ducts, and be minimally invasive or even noninvasive due to their small size [1]. Therefore, researchers apply microrobots to biomedical such as minimally invasive surgery, targeted drug delivery, and in situ biopsies [2,3,4,5,6,7,8]. Biomedical microrobots can effectively improve diagnostic or therapeutic capabilities, reduce damage and side effects during treatment, and promise new diagnostic or therapeutic techniques in the future [9].

The micromachines applied to non-invasive medical interventions have attracted numerous attention since being proposed by Albert Hibbs and Richard Feynman in 1959. After decades of development, it has made great progress in a variety of microrobot fields, including actuation, control, navigation, and functionalization, laying the foundation for future clinical applications [1,3,5,9,10,11,12,13,14,15,16,17,18,19]. However, how to accurately deliver microrobots to a specified position to perform medical tasks is still an unsolved problem due to the fact that manual control is inefficient and imprecise. Therefore, the autonomous actuation of microrobots is considered the most promising technology to solve the problem, in which operators give tasks and microrobots autonomously advance to the target position under the control of the host computer.

A microrobot’s autonomous actuation system (AAS) consists of three main components, namely actuation, control, and perceptive system, which is similar to traditional AAS. However, differently from the traditional autonomous actuation perceptive systems based on sensor feedback, researchers use medical imaging as the perceptive system to localize microrobots considering their small size and complex working environment in clinical applications [20].

There has been a lot of research on three main components of AAS for microrobots including actuation and medical imaging systems using various energy and control systems with different algorithms (Figure 1). The focus of current research is on the system integration to autonomously deliver microrobots to the target position in vivo. An integrated AAS platform is of great significance for functional microrobot applications in the body and has good performance. It can efficiently transport microrobots to perform tasks, reduce extra operators, and avoid injury for patients. In this paper, we investigated microrobot’s actuation systems, imaging systems, the integration of actuation and imaging systems, and autonomous actuation control systems, hoping to realize clinical applications in vivo by integrating AAS.

## 2. Microrobots’s Actuation

The actuation of the microrobots requires input energy, which is converted into mechanical energy, and the conversion usually requires special materials or structural design. There were two typical actuation models according to the input method: external field actuation and fuel-dependent self-propulsion. Given the self-propulsion method’s lack of effective control, we focused on the discussion of the external field actuation in this paper, which can be divided into magnetic, light, sound, and electric field actuation depending on the different energy sources. Furthermore, hybrid actuation supplying multiple actuation modes including hybrid external fields actuation and self-propulsion combined with external field actuation had improved actuation ability [10,21]. The mainstream actuation technologies are summarized in Table 1.

### 2.1. Magnetic Field Actuation

Magnetic fields acting on the microrobot consisting of magnetic material can be used to sense forces to actuate microrobot to realize magnetic field actuation. The external magnetic field applied can be divided into two main types: gradient magnetic field, which produces traction motion, and rotating magnetic field, which leads to spiral motion. Moreover, the rotating magnetic field is used for effective long-distance motion, while the gradient field actuation, which is more gentle, is suitable for fine operation such as direction alignment operation, so these two different magnetic fields often work together to realize actuation [15,22,23,24,25,26]. The magnetic field actuation system relies on both the arrangement of the coil and the current flowing through the coil. According to Faraday’s law of electromagnetic induction, the corresponding desired magnetic field distribution can be determined through the chosen controller and actuation system. A gradient magnetic field can be generated by reverse currents in a pair of coils. On the contrary, the rotating magnetic field is generated by alternating currents using multiple winding coils. The advantage of magnetic field actuation is that the human body is transparent to low-frequency magnetic fields, leading to higher actuation efficiency. However, the control magnetic field is applied to the whole body, which is difficult for decentralized control.

Zhang et al. [15] used a rotating magnetic field generated by three orthogonal electromagnetic coils to actuate the artificial bacterial flagella (ABF) by adding a soft magnetic head consisting of Cr/Ni/Au films on it. The magnetized soft head promoted the ABF forward under the action of the external rotating magnetic field, but this movement can only be carried out on one plane (Figure 2A). Zhang et al. [27] developed an electromagnetic actuation system (EMA) consisting of three pairs of Maxwell coils and three pairs of Helmholtz coils (Figure 2B). By independently altering the current in each coil, the EMA system can produce magnetic fields with different characteristics in three-dimensional (3D) space, such as controlled uniform gradient magnetic fields and rotating and oscillating magnetic fields. With this EMA system, the author developed a new wireless microrobot gravity compensation control method that used cylindrical magnets as control objects, which performed well for microrobots’ horizontal motion as well as various motions in 3D space. Zheng et al. [16] developed an octagonal electromagnetic system consisting of eight electromagnetic coils with DT4 cores distributing diagonally, which can generate gradient magnetic fields by applying the reverse current into adjacent coils, while rotating magnetic fields were caused by currents flowing through corresponding coils according to magnetic components in three directions. Meanwhile, the patterned seaweed hydrogels and magnetic microspheres were used as control objects. Consequently, the last two electromagnetic systems can achieve 3D motion control. In addition to the common gradient and rotating magnetic fields, oscillating magnetic fields were commonly used in microrobots with special structures, such as fins [28] and flat whiplash [29].

### 2.2. Light Field Actuation

Methods of light field actuation can be divided into the following four types: (1) The first method is tuation by the phototaxis of some microorganisms (the simplest and most direct method) [30]. (2) The second method is light radiation pressure to advance. Even though the light pressure actuation has no special requirements for microrobot materials, the higher radiation pressure of lasers was preferred compared with ordinary light sources [31,32]. (3) The third method is using light to produce a gradient field. Under the gradient field, the microrobot moves forward, for example, by photothermal swimming [33] and photo-induced electrophoresis [34]. (4) The fourth method is an additional actuator onthe microrobot for optical actuation such as photothermal actuation [31,35,36] and photoelectric actuation [37]. The advantage of light field actuation is that the light field is highly spatially selective and can control multiple microrobots independently at the same time [38].

Nagai et al. [30] used the phototaxis of phototactic algae as an actuation method. The green LED of a maximum luminous center at 525 nm illuminated from one side can induce the phototactic algae migration (Figure 3A). Avci et al. [11] actuated hinged microrobots directly utilizing light radiation pressure with a commercial optical tweezer (Elliot Scientific, E3500) as the power source. In the experiment, they fixed one component and rotated the other one, enabling the microrobot to move forward (Figure 3B). Compared with single rigid body end effectors, hinged microrobots can generate out-of-plane rotational motions, promising complex biomedical applications in 3D space. Li et al. [13] added an Au coating to the microrobot. When the near-infrared laser increased the tail temperature significantly the corresponding asymmetric temperature gradient pushed the microbar robot forward (Figure 3C). Benefiting from the multiple tubular channels, the micro-rocket can reach a speed approximately 3 times faster than the micro-tube robot and 7.5 times faster than the micro-rod robot. Zhang et al. [36] developed a new MEMS microrobot on SOI wafers, and the photothermal effect coming from the laser actuated the microrobot by combining sticky motion gait. Most existing microrobots have been simple, so they can only be controlled externally, and mass manufacture has relied on chemical synthesis with only some single specific function, which made it difficult to meet the needs of future complex medical tasks. Recently, Miskin et al. [37] used traditional silicon processing technology to fabricate microrobots actuated by light pulses that integrated photovoltaic cells for photovoltaic conversion and platinum electrodes as electrochemical actuators. More than one million microrobots can be produced on a four-inch wafer with a yield of roughly 9%, and different functional devices can be easily extended (Figure 3D). The CMOS process is inexpensive for mass manufacturing, and a single microrobot manufactured by the 180 nm process cost as little as $0.001. This mass-manufactured and integrated microrobot may be widely used in future complex medical applications.

### 2.3. Acoustic Field Actuation

Acoustic field actuation relies on the radiation of sound waves on the microrobot. Sound waves that travel through the fluid produce forward pressure to actuate the microrobot. The moving direction is the same as the sound propagation. The sound waves for actuation can be generated by the piezoelectric transducer and come in the form of surface sound, ultrasonic, ultrasonic standing wave, and pulsed ultrasound [39,40,41,42]. The advantages of acoustic field actuation are strong tissue penetration, high adjustability, and biocompatibility within the MHz range [43].

Aghakhani et al. [44] designed an acoustically actuated surface sliding microrobot that contained spherical bubbles trapped in its internal body cavity, which resonated with sound waves. The sound waves for actuation had a frequency of 330 kHz and depth between 2 to 20 cm, which were produced by a piezoelectric transducer with a radius of 1 to 3 cm (Figure 4A). The flow of clean fluid caused by bubble oscillations led the microrobot to move at high speed. Compared with microorganisms, the proposed microrobot had a thrust force of 2 to 3 orders of magnitude higher, which is sufficient for navigation inside vascular capillaries with blood flow. Ahmed et al. [39] electrochemically deposited single metal and bimetallic nanorods containing Au, Rh, Pd, Ag, Pt, or Ru chain segments. Ultrasonic waves were also generated by a piezoelectric transducer but reflected by glass cover slides to form standing waves for actuation (Figure 4B). Kaynak et al. [45] designed an acoustically actuated bionic microswimmer and used signal generators, RF power amplifiers, and piezoelectric transducer to generate sound waves (Figure 4C). The speed of the microswimmer was a function of peak-to-peak voltage (VPP) applied to the transducer. The linear microswimmers reached 1200 μm/s, while rotational microswimmers rotated at 200 RPM.

### 2.4. Electric Field Actuation

Electric field actuation relies on the Coulomb force applied on the charged object, so the microrobot can move in the direction of an electrode with opposite charges. Therefore, the microrobot was composed of charged materials or materials that can generate charges realized to be actuated remotely by applying DC or AC fields externally [46]. The actuation mechanism can be divided into asymmetric charge or asymmetrical shape-induced asymmetrical flow [47,48] and electroosmotic flow caused by local rectification [49]. But the effect of the electric field decayed rapidly with distance. Moreover, it is difficult to generate the required electric field in some high-ion media such as tissue fluids or blood, which limited the clinical applications [1].

Loget et al. [50] designed a dynamic bipolar self-regeneration actuated swimmer with different redox reactions at two ends of the substrate. Therefore, under the influence of external electric field, the metal deposited on one end of the swimmer and dissolved on the other end made it move forward (Figure 5A). The actuation mechanism can be extended to the nanoscale. Chang et al. [49] proposed a remotely powered self-propelled particle and micropump based on a micro diode, which was actuated by an external AC field. It can sense the DC voltage internally, which causes localized electroosmotic flow (Figure 5B). Meanwhile, the DC voltage could supply power to the internal logic circuit to realize the dynamic reconfigurable microfluidic chip and the spatially evolved active microsensor network.

### 2.5. Hybrid Actuation

The single-source actuation mode has different limitations, for example, the high decay over distances of magnetic field actuation, heat effect in acoustic field actuation, poor light penetration for light field actuation, and so on, making it difficult for it to cope with the complex environment and medical tasks in future clinical applications [1]. Hence, hybrid actuation, converting multiple external energy sources into the actuation power for one microrobot, can combine advantages of different actuation modes and further improve the microrobot motion performance and function in complex environments [51,52,53].

Jiang et al. [54] designed a dual-actuated bionic microrobot by depositing Au and Ni nanoparticles on the microrobot surface, such that Au and Ni nanoparticles provided photothermal and magnetic field actuation, respectively. The electromagnetic coils applied gradient magnetic fields for macro-manipulation, while infrared lasers supplied external light to tune microrobots in predetermined direction and position (Figure 6A). Li et al. [55] designed a magnetically/acoustically hybrid actuated nanomotor, which included magnetic helix structures and concave nanorod ends. It provided precise space–time control and regulation, leading to a considerable adaptive performance in different dynamically changing environments (Figure 6B). In addition to external field actuation, self-propelled microrobots have received great attention due to their high speed of movement. Self-propelled microrobots used catalytic reactions or transformed microorganisms/cells to convert biochemical energy into mechanical energy to realize actuation; however, this method lacked effective control [56,57,58]. Researchers combined self-propulsion and external field actuation to improve the control capability of the microrobot. Gao et al. [51] developed a catalytically/magnetically powered hybrid nanomotor. The Pt-Au and Au-$Ag_{flex}$-Ni parts played the role of catalyzing self-propelled and magnetic field actuation modes, respectively. The hybrid actuation had little impact on each mode function and could switch from catalytic to magnetic mode quickly and easily. In addition, the applied magnetic field could reverse the direction of movement (Figure 6C). Li et al. [52] designed a microrobot composed of beads containing magnetic particles attached by flagella bacteria. The microrobot can be self-propelled by flagella bacteria or actuated by an external electromagnetic system. The electromagnetic system can overcome the high-speed blood flow of large vessels and aims for accurate direction and position control. Then, in a small blood vessel, the microrobot can be delivered to a specific tumor area through the chemotaxis and motility of bacteria (Figure 6D).

## 3. Microrobot Imaging

A microrobot’s imaging system, the basis for its localizing and navigating, is critical in AAS. In order to track the microrobot in vivo, several important parameters of the imaging system, such as the imaging depth in vivo, imaging time, and resolution, have attracted great attention. However, conventional optical imaging techniques are difficult use in vivo, so researchers have combined medical imaging technologies to develop microrobot imaging systems for clinical applications. Considering the different energies used, medical imaging can be divided into magnetic field-based, optical, ultrasound, ionizing radiation, photoacoustic, and multimodal imaging. In principle, the energy is emitted to the human body and interacts with the environment in vivo (reflected or absorbed), and corresponding signals are processed to obtain the image. According to various characteristics in the image, such as the contour, intensity, and so on, the microrobot can be localized and tracked.The medical imaging technologies used to locate and track micro-robots are summarized in Table 2.

### 3.1. Magnetic Field-Based Imaging

Magnetic-field-based imaging has been widely used in medical diagnostic imaging by applying an oscillating magnetic field into the human body. The frequency and phase of returned signals reflects the internal tissue information. Therefore, the image can be generated by reconstruction. The magnetic field for imaging has a high-penetration depth and is little harmful to the human body, so it can be used for deep tissue imaging [59,60,61]. There are two conventional magnetic-field-based imaging modes: magnetic resonance imaging (MRI) and magnetic particle imaging (MPI).

MRI is an imaging technique developed from the 1970s to the 1980s. It uses magnetic resonance to obtain electromagnetic signals and reconstructs imaging information, with excellent tissue contrast, high spatial resolution, and deep penetration in a non-invasive and accurate manner [62]. The MRI equipment consists of a main magnet and gradient coil, which produces a highly uniform static and gradient magnetic field, respectively. In addition, a transmitter coil combined with an RF generator can generate RF pulses, while a receiver collects signals for image reconstruction. Firstly, the static magnetic field and gradient magnetic field are used to align the magnetic core spin. Then, the RF pulse causes a disturbance to this alignment as the nuclear spin absorbs the electromagnetic pulse corresponding to the resonance frequency, which is related to the magnetic field intensity. The energy returned during relaxation can be detected by the external receiver and depict the NMR spectrum. The H-core imaging is abundant in the human body, especially in fat and water environments, leading to its widespread clinical applications [63].

Olamaei and Dahmen et al. [64,65,66] used artifacts of magnetic microrobot in MRI images for localization, whose extent is related to scan parameters and particle size. Additionally, magnetic microbots can be controlled by an MRI system that has been upgraded to a custom gradient coil (Figure 7C). Felfoul et al. [12] have developed a new magnetic resonance tracking method based on a selective excitation ferromagnetic core. With RF signal tuning, the position signal of the ferromagnetic core can be seen clearly in the image.

Magnetic particle inspection (MPI) is a new tomography technology that can image the magnetic particles’ space distribution with high spatial resolution. MPI is based on the nonlinear response of magnetic particles to magnetic fields, meaning that magnetic particles have a harmonic response to the alternating magnetic field at field-free points. The corresponding intensity is proportional to particles’ concentration, and magnetic particles have little response in the saturated magnetic field area. Hence, through constructing field-free points in space and applying an alternating magnetic field, the harmonic signal measured can illustrate the distribution of magnetic particles [67]. MPI equipment includes a selection coil for generating field-free points, a drive coil for producing an alternating magnetic field, a measurement coil for signal detection, and a computer for image reconstruction. MPI has faster image speed and lower cost compared to MRI.

MPI is an imaging method based on magnetic particles that can be actuated and functionally targeted for microrobots in drug delivery and situ sensing applications [68,69]. Le et al. [70] have designed a real-time 2D MPI for electromagnetic navigation, which monitored Resovist particles at 45 to 65 nm (5 nm core) in real-time (Figure 7D).

### 3.2. Optical Imaging

The medical application of optical imaging is in one-sided optical pathways for the strong light absorption in the body. There are two optical imaging forms based on reflection and fluorescence. The conventional optical imaging equipment includes microscopes (confocal, fluorescent, and polyphon fluorescent microscope), optical coherence tomography (OCT), endoscope, and so on [71]. The optical imaging can reach a high resolution, but the penetration depth is weak.

Li and Wu et al. [6,72] used OCT imaging as feedback to enable autonomous navigation of magnetic field actuated microrobots in vivo (Figure 8A). OCT has good depth-recognition capabilities, which can be compared to Dual photon fluorescent microscopes and confocal microscopes. However, the point-by-point data capturing leads to low imaging speed. Yang and Xing et al. [73,74] performed fluorescence imaging tracking magnetic biomicrorobot coated by fluorescent agents. The imaging system consists of a fluorescent microscope, fluorescent illuminator, and CMOS camera, in which fluorescent agents are excited under fluorescent illuminators (Figure 8B). Aziz et al. [75] designed single reflective micromotors by depositing magnetic layers on silica particles for external magnetic field control and highly reflective thin gold layers as a micro reflector. An imaging system, consisting of a 970 nm infrared LED as a light source, microscope, and infrared camera, was used for micromotor real-time tracking in the tissue (Figure 8C). Wang et al. [76] have designed an integrated platform for endoscope-assisted magnetic actuation dual imaging systems (EMADIS) that allowed magnetic actuation microrobot to be tracked in real-time through the endoscope view after injection (Figure 8D).

### 3.3. Ultrasound Imaging

Ultrasound (US) imaging is widely used clinically, including B-mode, and Doppler imaging in common. Better technology like harmonic can further improve image quality [77]. Ultrasonic imaging equipment is mainly based on piezoelectric transducer arrays, which are responsible for electro-acoustic conversion. The ultrasonic wave produced by the transducer array is scattered when it encounters sound impedance discontinuity in the body. The scattered echo signals can be detected by the transducer array, leading to obtaining images. In Mode B, the scattered echo signal is reflected as grayscale, and the value illustrates the reflection intensity and attenuation of ultrasonic waves at the echo interface. The Doppler technology, relying on detecting the changed frequency of ultrasonic waves after interacting with moving objects, can be used to obtain dynamic images, such as application in doppler flow detection. Pulses are emitted at one frequency and the receiver received transmit frequency at twice contrast in harmonic imaging [78,79]. The greatest advantage of US imaging is the high penetration depth, implying application for localizing microrobots in deep tissues.

There have been many studies using US imaging directly as feedback to control a single microrobot without additional operations [80,81,82]. The microrobot needs to be big enough to be observed at submillimeter resolution for US imaging. However, the signal coming from a small-size microrobot was too weak to be obtained. Wang et al. [83,84] used US imaging to track a nanoparticle microswarm, which exhibited enhanced US imaging compared with individual nanoparticles (Figure 9). In addition to directly tracking the microrobot, US imaging can indirectly track microrobot by imaging microbubbles generated by microrobot actuation. Sánchez et al. [85] designed a microtube made of titanium, iron, and platinum nanofilms. The microtube can be self-propelled by the catalysis of the platinum nanofilm on H_2_O_2_, and the applied magnetic fields can guide it toward the intended target. Therefore, the microbubbles created during self-propulsion were observed in ultrasonic images with high quality due to the strong scattering effect of microbubbles to ultrasonic waves.

### 3.4. Ionizing Radiation Imaging

Ionizing radiation imaging utilizes high-energy radiation to penetrate the human body, and the commonly used clinically ionizing radiation imaging technologies include CT, PET, and SPECT. CT irradiates the human body with X-rays, which attenuate depending on different densities and thicknesses of tissues or organs in the body, thus forming a comparative distribution of gray-scale images. Both PET and SPECT require injecting imaging agents and use γ rays for imaging. SPECT uses radionuclides that directly emit γ rays for imaging, while in PET, radionuclide images decay by emitting positrons, which bind to the electrons to produce γ rays after traveling a short distance through the surrounding tissue [86]. Ionizing radiation imaging has high spatial resolution, but the radiation can be harmful to human tissue if the applied dose is too large or lasts a long time. In addition, it is important that CT reflects anatomical images, while PET and SPECT dynamically report the body’s metabolic information instead of anatomical details.

Vilela et al. [87] used PET to precisely track the position of micromotors with chemisorption of the iodine isotope onto the Au layer, providing quantitative spatial information in the body (Figure 10A). The PET-CT imaging was consistent with optical microscopes observation. Nguyen et al. [88] filled the magnetically guided microrobot with X-ray contrast agents to be pinpointed in real-time. X-ray imaging and the microrobot can be loaded with therapeutic agents for precise drug delivery systems using autonomous control (Figure 10B).

### 3.5. Photoacoustic Imaging

Photoacoustic imaging (PAI) is a new biomedical imaging method based on the thermoelastic expansion of molecules after absorbing light to produce ultrasound. The nanosecond pulsed laser (pulse duration < 10 ns) is used to illuminate biological samples, and the ultrasound can be detected by the ultrasonic transducer array with high optical contrast and spatial resolution. Since the scattering of sound is 1000 times less than that of light, acoustic signals can travel further in biological tissues, leading to deeper imaging depth [89,90].

Previous studies used PAI to track microrobots by adding contrast agents to microrobots. The contrast agents, including metal nanoparticles, carbon-based nanomaterials, quantum dots, small organic molecules, semiconductor polymer nanoparticles, etc., can provide strong contrast due to the high absorption of light in some optical windows [91]. Moreover, the contrast agents on the microrobot can enable other functions. For example, nickel (Ni) particles can be magnetically actuated without adding additional materials (Figure 11A) [92,93], and polydopamine (PDA) can realize diagnostic sensing by further modifying through diverse surface reactions (Figure 11B) [94].

### 3.6. Multimodal Imaging

Different single imaging modes have limitations, making it difficult to meet the need to localize and track microrobots in the complicated environment of the human body [20,63]. Multimodal imaging integrating two or more complementary methods can gather information, improve the image localizing effect, and provide additional structure, function, dynamics, or molecular composition information, providing more possible clinical applications of microrobots [95,96,97]. A variety of multimodal imaging options have been studied, such as PET/MRI multimodal imaging with excellent soft-tissue contrast and precise localization within organs [97], and the combination of PAI and OCT with ultra-high resolution for obtaining valuable vascular depth information [98]. Yan et al. [92,93,99] designed an integrated ultrasound and photoacoustic (USPA) imaging system (Figure 11A) to track magnetically manipulated microrobots in deep tissues, combining advantages of US and PA. The US imaging visualized the anatomical and structural information of tissues, and PA imaging increased sensitivity to detect microscale objects. In addition, Vilela et al. [87] used PET-CT to navigate micromotors (Figure 10A) to effectively overcome the lack of images in the working space for PET navigation. Moreover, researchers have developed contrast agents for multimodal imaging which can be used to localize and track microrobots [100].

## 4. Actuation and Imaging Integration

To realize actuating in vivo autonomously, it needs to integrate both actuation and imaging into a system controlled by the host computer. The first two chapters have independently summarized the existing actuation and imaging methods. For actuation, a set of transmitting equipment was necessary, while additional receiving equipment was required for imaging. Therefore, it considers whether to share the transmitting equipment for the integrated system, which depends on whether the energy used for imaging and actuation is the same. In addition, it is necessary to consider some requirements according to actuation and imaging needs. On the one hand, external actuation and imaging equipment do not interfere with each other and can be arranged together in a limited space. On the other hand, appropriate materials are usually added on microrobot to respond to external controls or as contrast agents for localizing and tracking in medical images. According to sharing transmission equipment or not, the actuation and imaging systems integration was divided into two situations, in which appropriate methods are chosen according to actual needs and features of different methods (Table 3).

### 4.1. Shared Transmission Equipment

The shared transmission equipment for both imaging and actuation is arrangeable and low-cost when the energy used is the same. Moreover, existing medical imaging equipment is well-established, whose transmission equipment can be upgraded to actuate microrobots with the reduction of equipment cost and complexity.

Existing magnetic field-based imaging equipment (MRI and MPI) contains multiple electromagnetic systems that can be upgraded to actuate microrobots [101,102,103,104]. The magnetic field actuation and imaging methods have been widely studied because the magnetic field has deep penetration depth and little harm. It can be used for deep tissue imaging and actuation with good control. Belharet et al. [105] have designed an MRI-based miniature robotic system that can actuate magnetic microcapsules or nanocapsules in cardiovascular systems. MRI coils were used to generate magnetic gradients and magnetic fields to sense forces and torques, respectively. Griese et al. [106] used MPI equipment to image and actuate magnetic beads for targeted delivery with 7 Hz time resolution.

In addition to the magnetic field, there have been other imaging and actuation integration methods that have applied the same energy. For example, Li et al. [13] designed a micro-rocket robot with all-optic actuating and tracking. The robot was actuated by the photothermal mechanism, while the photoacoustic microscope was used to track individual microrobots in the blood with high resolution. Notably, the coated Au on the microrobot surface can be used as actuation material and PAI contrast agent at the same time. Xu et al. [107] utilized X-ray for actuation and real-time visualization of Janus particles, which caused the radiolysis of water around the particle, creating bubbles to actuate the Janus particle.

### 4.2. Independent Transmission Equipment

To avoid interference between the actuation and imaging system when sharing transmission, a time-sharing method is required. It means only one mode can be performed at a time and often needs to be switched; therefore the efficiency is low. In addition, the selectivity of actuation and imaging equipment is another important consideration.

Consequently, researchers have used independent transmission equipment for actuation and imaging to improve performance. Taking the most widely studied electromagnetically actuated microrobot as an example, in addition to magnetic field-based MRI and MPI imaging methods, optical and ionizing radiation and US imaging were combined with electromagnetic actuation systems for image-guided navigation of microrobots [88,108,109,110]. To solve the problem that the imaging resolution of MRI was insufficient to localize microrobots smaller than 100 μm, Li et al. [72] designed a microrobot navigation platform consisting of an electromagnetic actuation system and an OCT imaging system that can track microrobot movement at high scanning rates (5.5–70 kHz) and resolution (10 μm). Other researchers used US imaging to track and image electromagnetically actuated magnetic particle populations, simply adding ultrasonic probes to electromagnetic actuation systems. The equipment was simple, with low cost, deep imaging depth, and high time resolution [83,84]. Wei et al. [111] used PA and US dual-mode imaging systems to guide clusters of gilded microrobots, which were actuated by gradient magnetic fields with a clear image at a depth of 2 cm.

The actuation and imaging systems using independent transmission equipment were more selective, allowing different imaging/actuation methods to be used depending on the requirements. However, it needed additional equipment costs, placement, interference, and so on. In addition, extra materials were added on microrobot to respond to external controls and increase image contrast for localizing and tracking in medical images.

## 5. Microrobot’s Medical Imaging-Based Autonomous Actuation Control

The previous discussions were of the hardware aspect of AAS. In order to realize autonomous actuation, the software cooperating with the hardware equipment is also important. The control system is the core part of the software, in which images are processed after the imaging system obtaining information and the microrobot is navigated according to specific characteristics, and then instructions are sent to the actuation system to control the microrobot motion. Similar to the traditional autonomous control system, there are two main tasks: planning the path of the microrobot according to external instructions and imaging guidance along the planned path to correct the trajectory of the microrobot.

### 5.1. Path Planning

The purpose of path planning is to find an optimal path between the start point and the endpoint, which refers to the current position of the microrobot and the target position set by the operator in the microrobot navigation, respectively. According to Fernando Soto, the main ways microrobots enter the human body are through the mouth and intravenous injection [112]. Both of them transmit relying on the human body’s vascular network, so it is important to plan pathways within the vascular network. There are two aspects to consider: (1)extracting the vascular pathway from the medical image. and (2) planning the path. The extraction of vascular images has made great progress according to computer vision and medical imaging [113], on which microrobot researchers have planned paths.

Belharet et al. [105,114,115] have used Frangi vascular filtration and Fast Marching Method (FMM) algorithms to extract blood vessels from preoperative images (3D MRI) and plan paths within the vascular network to navigate ferromagnetic microcapsules, respectively. The A-algorithm can generate paths quickly in different situations: low spatial dimensions, complex mazes, and narrow paths. However, paths are often too close to obstacles and can cause collision problems. Lim et al. [116] improved the A-algorithm to plan electromagnetically actuated microrobots path, which was generated by the improved algorithm with high autonomous actuation stability. Scheggi et al. [117] compared the path planning algorithms for six magnetic-field-actuation obstinate particles: A and Fork, A and Unified Grid, D-Lite, Artificial Potential Field, Probability Roadmap, and Quick Exploration Random Tree. In the given four cases, the six algorithms were equivalent in terms of path length and completion time. However, all results were done based on a single microrobot, which was not sufficient for clinical applications. Huang et al. [118] have proposed two problems: (1) the microrobot group was distributed in different parts of the vascular network and all units moved in the same direction with the same speed until obstacles were encountered, and (2) the microrobot group needs to be collected quickly at the aggregation point. The authors used an augmented RRT for trajectory generation to reduce environment interference. In addition, a divide-and-conquer algorithm for swarm aggregation can improve performance and be used as an effective control strategy for drug delivery in vascular networks.

### 5.2. Imaging-Guided Autonomous Actuation

The imaging guidance corrects the trajectory of the microrobot according to the error between the planned path and the current position obtained from the medical image, so that the microrobot can accurately reach the target position. To accomplish this task, a path was pre-planned firstly, and an imaging system was used to track the microrobot and obtain the position error, and then the actuation system was controlled to correct the trajectory of the microrobot by the corresponding controller.

Target tracking can be divided into two parts: feature extraction and target-tracking algorithms. For the microrobot in vivo, the main characteristics are intensity and contour. Nguyen et al. [119] developed an algorithm based on principal component analysis (PCA) to extract the tip and center point of the microrobot from X-ray images to obtain the position and direction of the microrobot. The processing time was less than 0.1 ms, which can be used to track the microrobot in vivo in real time. Yang et al. [73] used the different intensity between the fluorescent object and the background to track the fluorescent microrobot, with high resolution and strong contrast. After the feature extraction, the target tracking algorithms were used to localize the microrobot. Sharifi et al. [120] used the novel Kalman Filter (KF) algorithm to design the tracker to provide the best estimate of the trajectory of the microrobot. Zhang et al. [121] compared the tracking accuracy and time consumption of 12 tracking methods and chose the best-performing spatio-temporal context (STC) learning algorithm for real-time microrobot tracking.

After obtaining the real-time position of the microrobot, researchers designed different controllers for autonomous actuation of the microrobot. Kummer et al. [122] designed an electromagnetic system called OctoMag to control the microrobot, derived both current and force models on the microrobot, and used the position information obtained by visual localization as feedback for PD control. To cope with the pulsation pressure in the bloodstream, Choi et al. [123] used PID to accurately control the microrobot by generating an input that compensated for the drag force through the pressure sensor signal and obtaining the microrobot’s position through the CMOS camera. Both of these guidance schemes were based on using a camera and were difficult to apply in clinical applications. Belharet et al. [114] used 3D MRI to navigate ferromagnetic microrobots in blood vessels, and the model predictive control (MPC) could solve the problem of poor time resolution in MRI and external disturbances at work. The simulation results showed the effectiveness of the control algorithm. In addition to MRI, other medical imaging techniques can also be used in microrobot image guidance, such as fluorescence imaging [73] and X-ray imaging [124]. Considering 3D imaging reduced resolution or increased processing time, Barbot et al. [125] studied the orbital effect of rotating microrobot in channel operation; when a rotating robot was placed in a closed microchannel, the rotation created a force perpendicular to the channel directly on the robot, leading the robot to run around the centerline of the channel. Therefore, the authors suggested that navigation of 3D channels can be achieved with only 2D image feedback, which can greatly improve the performance of image guidance.

## 6. Conclusions and Outlook

In this paper, we investigated the various components of microrobots’ AAS, including actuation, medical imaging localization, actuation and imaging integration, and control. A lot of progress has been made for each component, but further efforts are still needed for an integrated AAS system that can be used in clinical applications. In this section, we conclude with the current progress and limitations of each component and discuss the challenges of future research and potential technical solutions.

### 6.1. Actuation

Most of the current actuation researches has concentrated on 2D planes and has lacked reliable dynamics and kinematic models, which make it difficult to meet the complex task requirements of clinical applications. Therefore, future researches should firstly focus on the actuation of 3D space. Secondly, reliable movement and steering control need to be established to provide precise control of microrobot motion in the body. Thirdly, hybrid actuation can carry out long-distance and fine actuation to achieve whole-body work [1]. Lastly, actuation and control of microrobot populations are urgently needed to conquer the limitations of the single microrobot in the body [126].

### 6.2. Imaging

Traditional medical imaging techniques have been widely used in clinical applications, but 3D localization for microrobot in real time and deep tissue and with high resolution is still difficult. Therefore, new imaging technologies are needed to improve microrobot’s image localization in the human body [20,127]. Recently, multimodal imaging has attracted more attention, combining advantages of different technologies to obtain inside information and track microscale objects for navigational control at the same time. In addition, multimode imaging also provides anatomy, morphology, function, and metabolism data in real-time, which is important for microrobot control and practical clinical applications [128].

### 6.3. Actuation and Imaging Integration

Research on the integration of actuation and imaging equipment is still in the early stages. Exiting method is single, and the actual environment is less considered. The next step will require further research based on actual needs and the design of external equipment. Furthermore, hybrid actuation and multimode imaging methods can be integrated with improved performance, as well as reusable and low-cost characteristics. In addition, the registration of the imaging and the motion space in the 3D state is still an unsolved problem.

### 6.4. Medical Imaging-Based Autonomous Navigation Control

Autonomous navigation control in the body relies on medical images as the information source. The previous research studies on path planning were mostly in 2D and used cameras for imaging. In the future, 3D path planning using medical imaging will be needed. At the same time, image guidance based on image feedback also requires further improvement to enhance microrobot control and deal with the complex situation within the human body, such as long medical imaging time, poor resolution, and interference in the actuation algorithm. In addition, 3D controls matching the actual motion space of the microrobot is another important problem to be solved.

### 6.5. System Integration for Autonomous Actuation In Vivo

The main challenge for AAS is creating a fast and high-precision in vivo imaging system, a high-performance actuation system, and a core autonomous actuation system. The challenge of integration lies in the placement of external devices, the efficiency of imaging and actuation, the mutual interference, the registration of the working space, the design of the microrobot, and so on. In addition, the interference caused by the working environment, such as the influence of blood flow on actuation, the contrast of imaging, etc., need to be studied. The ultimate goal is to realize an integrated AAS platform to provide a foundation for future medical applications in the human body.

The purpose of the entire AAS is to quickly deliver microrobots precisely to the target position in the human body for specific medical tasks. Due to the tiny size, microrobots can reach complex and narrow areas within the human body with high motion accuracy, which is not possible for traditional minimally invasive medical devices and surgical robots. However, according to current research, the microrobot movement speed was still insufficient considering the size of the human body. It was time-consuming to use microrobots for full-body-scale medical tasks. One possible approach was deliver microrobots rapidly by embedding catheters or directly injecting microrobots near the target area and then using AAS to preciselyactuate microrobots in complex and narrow circumstances [76,129].

## Figures and Tables

**Figure 1 micromachines-12-01249-f001:**
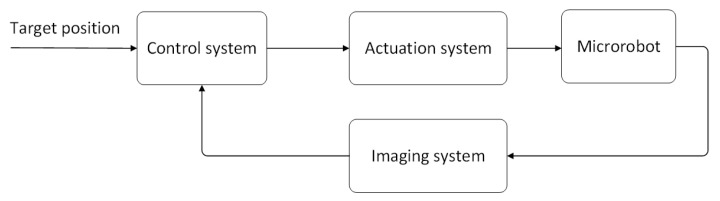
Block diagram of the AAS.

**Figure 2 micromachines-12-01249-f002:**
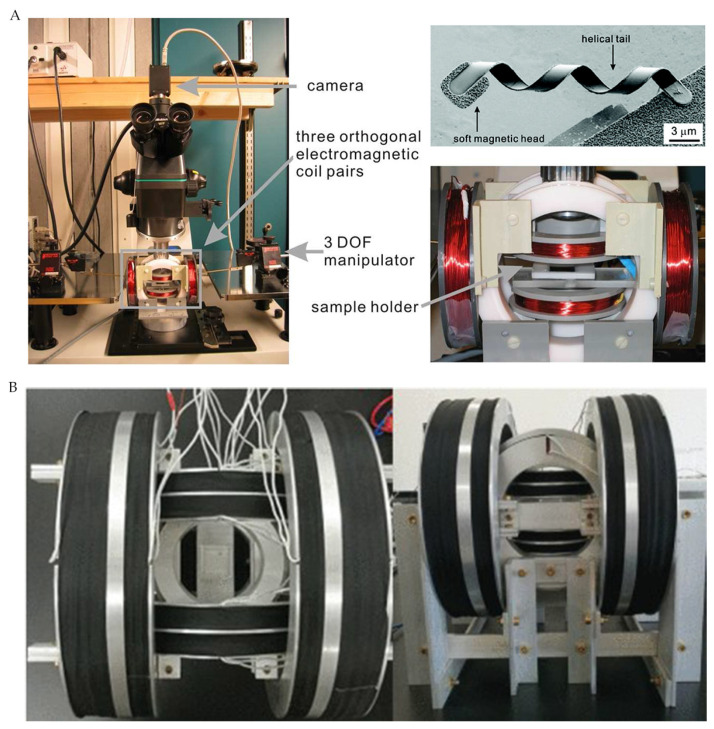
(**A**) Magnetic microswimmer and its electromagnetic actuation system. Adapted with permission from [15]. Copyright 2009 American Chemical Society. (**B**) EMA system. Reproduced with permission from ref [27]; published by IEEE, 2020.

**Figure 3 micromachines-12-01249-f003:**
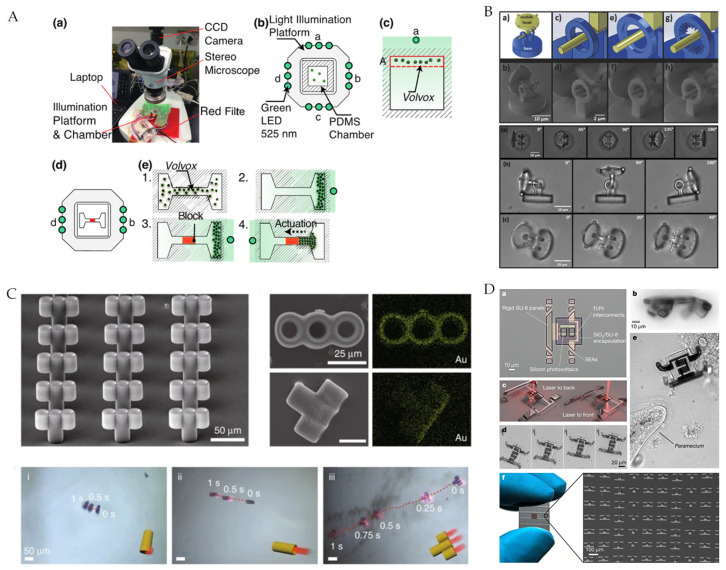
(**A**) Light-induced motion and equipment for phototactic algae. Reproduced with permission from [30]; published by MDPI, 2019. (**B**) Light-radiation pressure-actuated hinged microrobot. Reproduced with permission from [11]; published by John Wiley and Sons, 2017. (**C**) Photothermally actuated microbar robot. Reproduced with permission from [13]; published by Springer Nature, 2020. (**D**) Electronically integrated and mass-manufactured photoelectrically actuated microrobot. Reprinted by permission from ref [37]. Copyright 2020 Springer Nature.

**Figure 4 micromachines-12-01249-f004:**
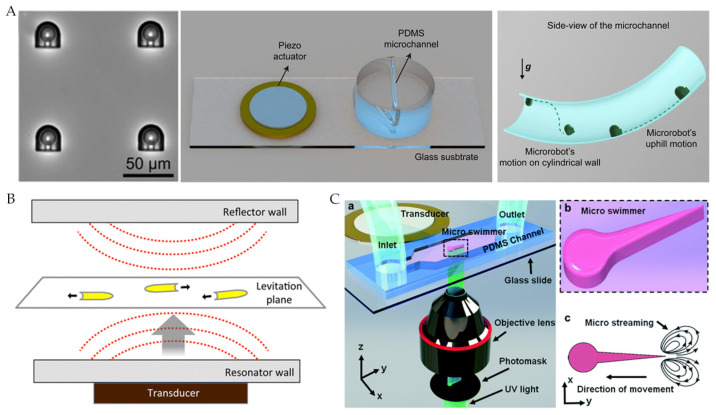
(**A**) Acoustic-actuated microrobot sliding on a 3D curved surface. Reproduced with permission from [44]; published by PNAS, 2020. (**B**) Standing waves actuated microrobot. Adapted with permission from [39]. Copyright 2016 American Chemical Society. (**C**) Acoustic-actuated bionic microswimmer. Reproduced with permission from [45]; published by Royal Society of Chemistry, 2017.

**Figure 5 micromachines-12-01249-f005:**
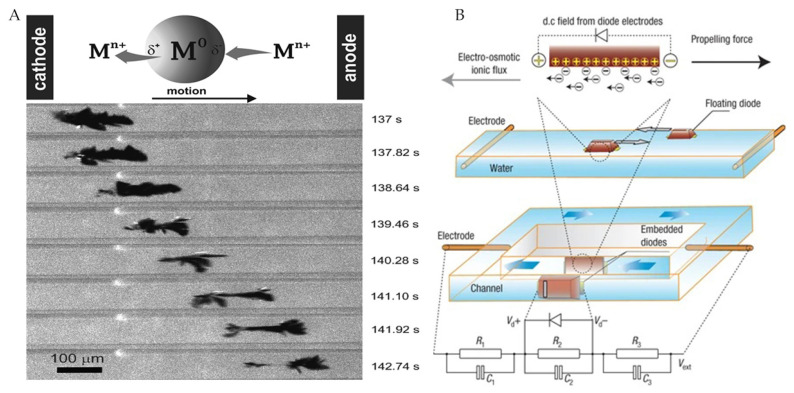
(**A**) Dynamic bipolar self-regeneration actuated swimmer. Adapted with permission from ref [50]. Copyright 2010 American Chemical Society. (**B**) Electroosmotic flow propulsion caused by local rectification. Reprinted with permission from [49]. Copyright 2007 Springer Nature.

**Figure 6 micromachines-12-01249-f006:**
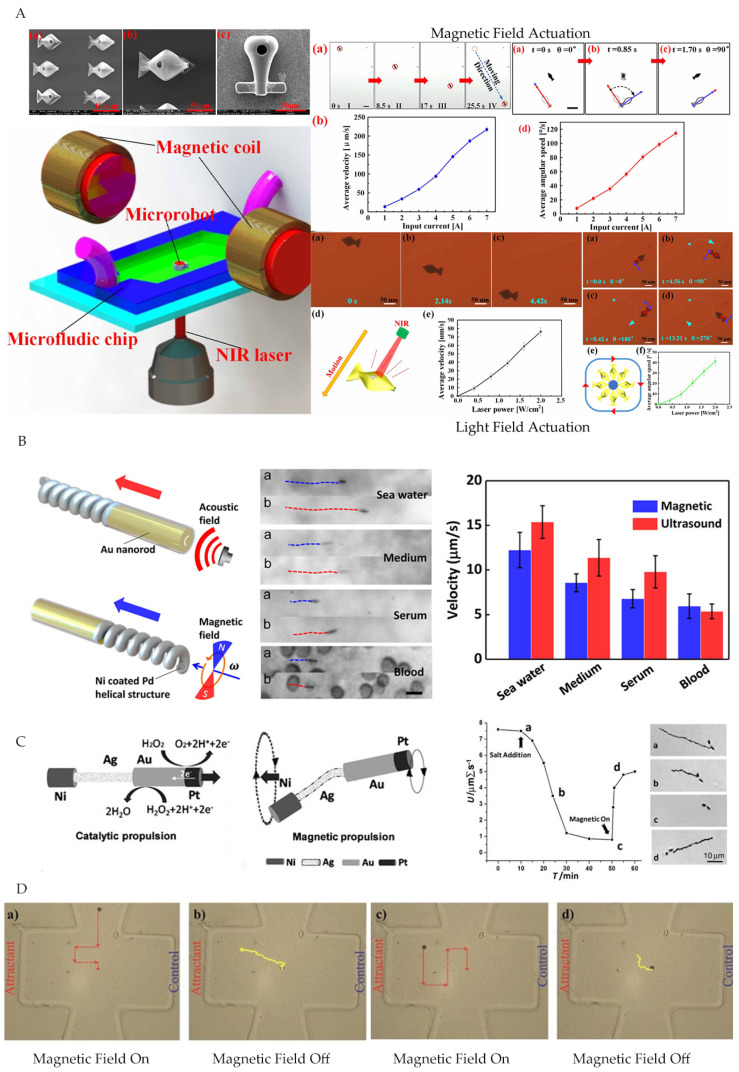
(**A**) Light/magnetic field hybrid actuated bionic microrobot. Reproduced with permission from ref [54]; published by Insitute of Physics Publishing, 2021. (**B**) Magnetically/acoustically hybrid actuated nanomotor [55]. Reprinted (adapted) with permission from ref [55]. Copyright 2015 American Chemical Society. (**C**) Catalytically/magnetically powered hybrid nanomotor. Reproduced with permission from [51]; published by John Wiley and Sons, 2011. (**D**) Microrobot actuated by hybrid magnetic field/flagellate bacteria. Reproduced with permission from ref [52]; published by John Wiley and Sons, 2015.

**Figure 7 micromachines-12-01249-f007:**
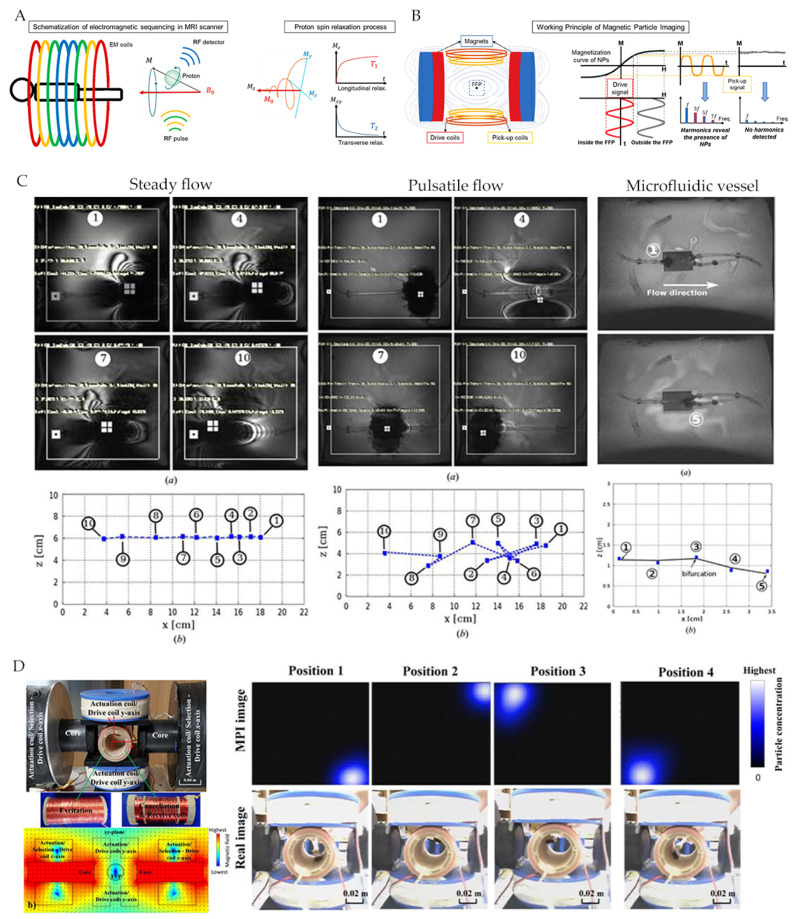
(**A**,**B**) are the working principles of MRI and MPI, respectively. Adapted with permission from [20]. Copyright 2020 American Chemical Society. (**C**) Localization based on magnetization artifacts in MRI images. Reproduced with permission from [66]; published by Taylor & Francis, 2016. (**D**) 2D MPI localizing. Reproduced with permission from [70]; published by MDPI, 2017.

**Figure 8 micromachines-12-01249-f008:**
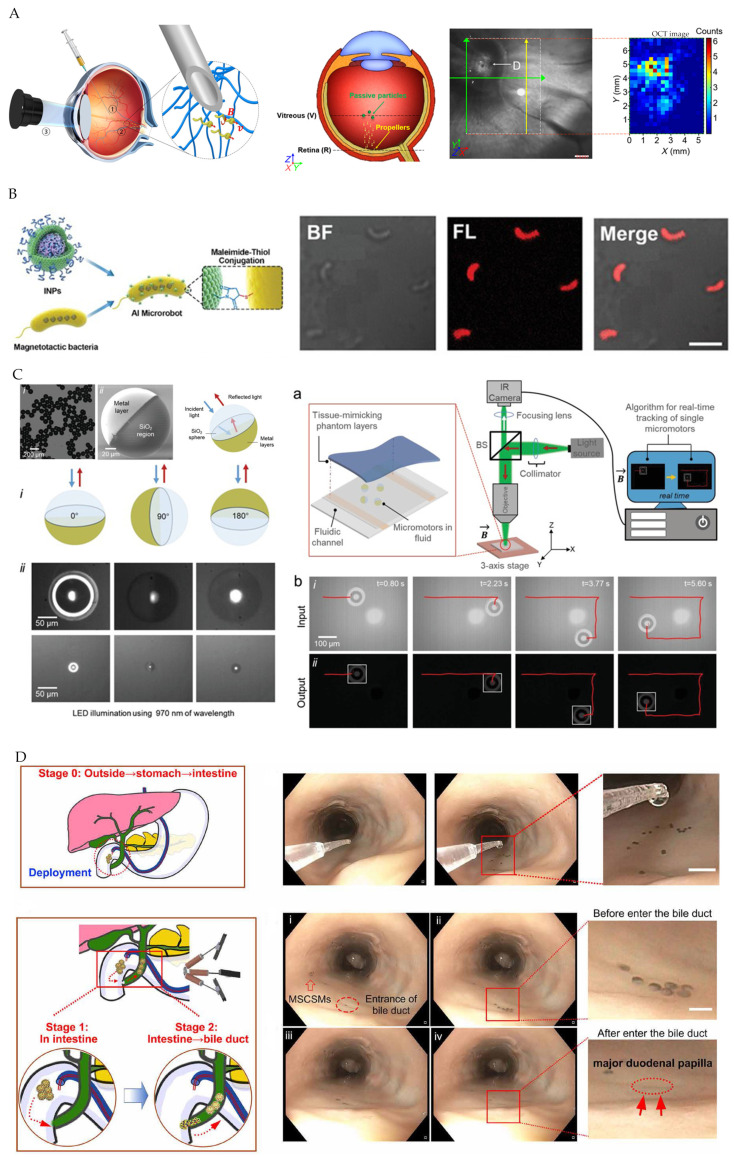
(**A**) Microrobot localized with OCT imaging. Reproduced with permission from [6]; published by AAAS, 2018. (**B**) Magnetic biomicrorobot tracked by fluorescence imaging. Reproduced with permission from [74]; published by John Wiley and Sons, 2021. (**C**) Single reflective micromotors tracked by real-time IR imaging. Reproduced with permission from [75]; published by John Wiley and Sons, 2019. (**D**) Endoscope-assisted tracking microrobot. Reproduced with permission from [76]; published by AAAS, 2021.

**Figure 9 micromachines-12-01249-f009:**
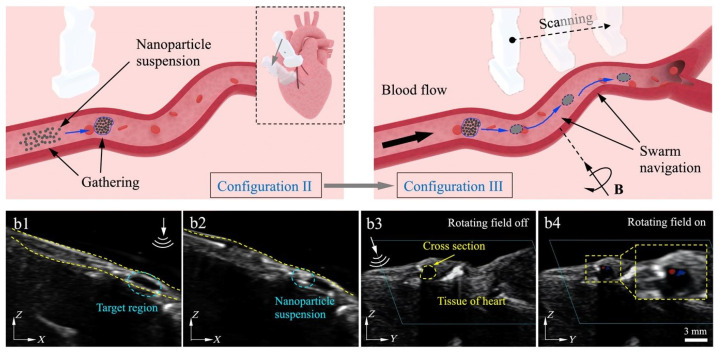
Magnetic microswarm tracked by US imaging. Reproduced with permission from ref [84]; published by AAAS, 2021.

**Figure 10 micromachines-12-01249-f010:**
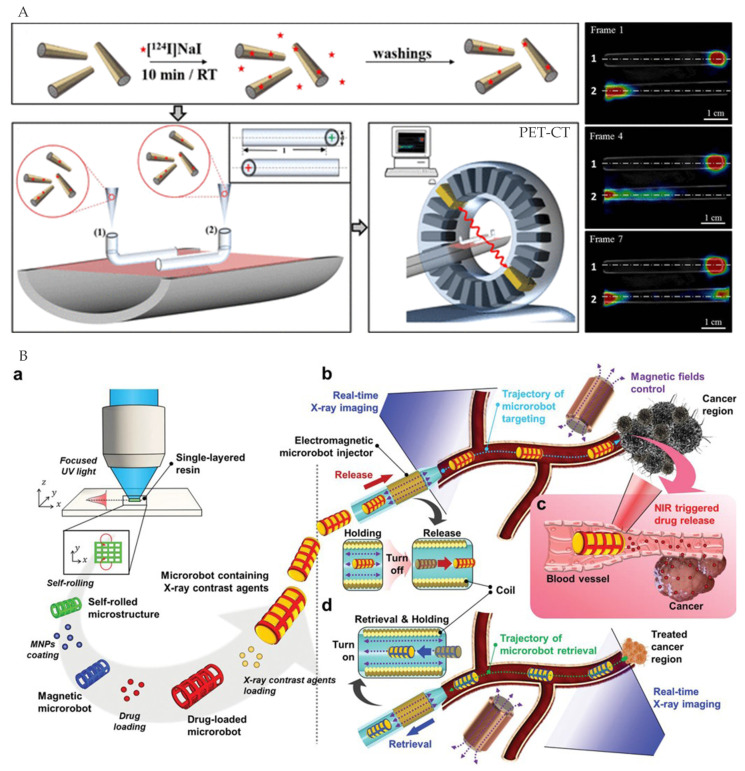
(**A**) Tracking by PET. Adapted with permission from [87]. Copyright 2018 American Chemical Society. (**B**) Tracking by X-ray imaging. Reproduced with permission from [88]; published by John Wiley and Sons, 2021.

**Figure 11 micromachines-12-01249-f011:**
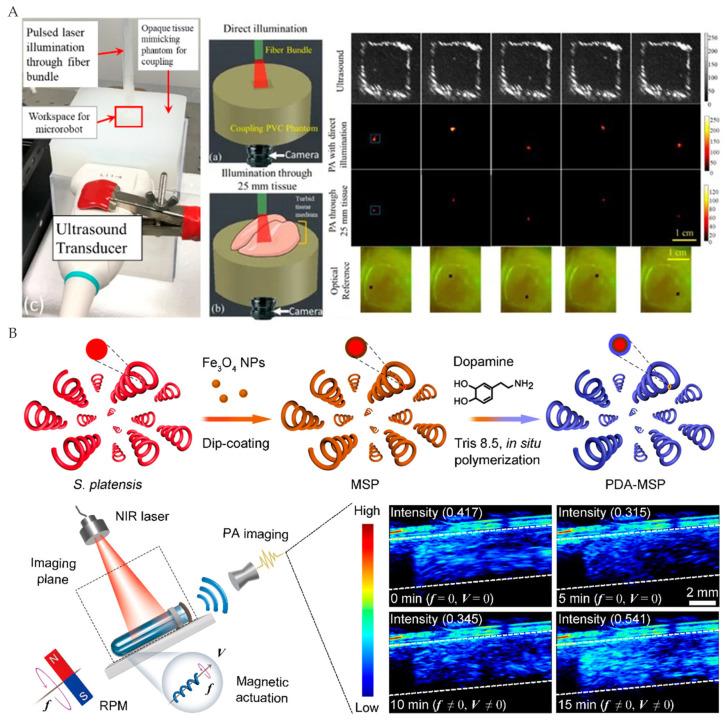
(**A**) Ni particles as PAI contrast agents. Reproduced with permission from [92]; published by MDPI, 2020. (**B**) PDA as PAI contrast agents. Adapted with permission from [94]. Copyright 2020 American Chemical Society.

**Table 1 micromachines-12-01249-t001:** Summary of mainstream actuation technologies.

Actuation Method	Actuation Equipment	Microrobot’s Feature Materials/ Structural	Microrobot’s Feature Size	Ref
Magnetic field	Three orthogonal electromagnetic coils	Nano-helix robot: InGaAs/GaAs/Cr helical tail and Cr/Ni/Au soft-magnetic head	Diameter 2.8 μm	[15]
Magnetic field	3 pairs of Maxwell coils and 3 pairs of Helmholtz coils	Cylindrical NdFeB (N42) magnet	2 mm in diameter and 3 mm in height	[27]
Magnetic field	Distributed diagonally in eight electromagnetic coils with DT4 cores	Patterned seaweed hydrogel with PEGDA magnetic microspheres	−500 μm	[16]
Light field	A green LED (emission maximum centered at λ = 525 nm, outer diameter: 3 mm, Stanley Electric, UG3803X, Tokyo, Japan)	Phototactic algae		[30]
Light field	Commercial optical tweezers (Elliot Scientific, E3500)	Hinged microrobot	−20 μm	[11]
Light field	808-nm laser with focus size of 354 μm	Au coating	45 μm	[13]
Light field	532 nm laser	Si	520 × 260 μm	[36]
Light field	Pulsed laser	Si/Pt	40 × 70 μm	[37]
Acoustics	Piezoelectric transducer	Au, Rh, Pd, Ag, Pt, and Ru metals	−1 μm	[39]
Electric field	Two electrodes diode		mm level	[49]
Hybrid actuation	Three-axis Helmholtz coil	Multi-segment Pt-Au-$Ag_{flex}$- Ni nanowires	10μm	[51]
Hybrid actuation	Infrared lasers and electromagnetic coils	Ni/Ain	−100 μm	[54]
Hybrid actuation	Electromagnetic coils and piezoelectric transducer	Nickel-plated gold nanorod and nickel-plated niobium nano helix	−50 μm	[55]

**Table 2 micromachines-12-01249-t002:** Medical imaging technology for localizing and tracking microrobots.

Imaging Method	Microrobot’s Feature Material	Microrobot’s Feature Size	Ref
MRI	Magnetic particles	−15 μm	[64,65]
MPI	Resovist particles	45 to 65 nm	[70]
OCT		Diameter 90 μm	[72]
Fluorescence imaging	Carbon point		[73]
Light reflection imaging	Silicon dioxide and a thin gold layer	100 and 20 microns	[75]
US	Rotate the colloidal group	A single 500 nm	[83]
US	Platinum microtubes		[85]
PET	Au/ Iodine isotope	−12 μm	[87]
CT	X-ray contrast agent	Radius 250 μm	[88]
PAI	Ni particles	>50 μm	[92]

**Table 3 micromachines-12-01249-t003:** Advantages and limitations of different methods.

Method	Actuation	Medical Imaging
Magnetic	High penetration depth; good actuation and control performance;safe; decays with distance	MRI: High penetration depth; good resolution; long imaging time. MPI: Fast imaging; limited field of view
Light	Low penetration depth; high spatial selectivity, which can be decentralized to control multiple microrobots;	High resolution; poor penetration depth
Acoustic	High penetration depth; safe; no special requirements for microrobot’s materials; heat generation when actuating	Fast imaging; poor resolution
Electric	Low cost; high-ion media required	
Radiation		High penetration depth; high resolution; long imaging time; hazardous
PA		Fast imaging; high resolution; improved penetration depth
Hybrid Actuation/ Multimodal Imaging	Higher actuation and control performance	Multi-dimensional information; better localizing performance

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
