# Peer review of "A Review of Microrobot’s System: Towards System Integration for Autonomous Actuation In Vivo"

_micromachines, 2021, doi:10.3390/mi12101249_

Round 1

Reviewer 1 Report

Microrobots are one of the very hot topics of the current material science and nanotechnology. The idea behind of these autonomous machines is that they behave as artificial microorganisms. The long-standing promise of this field is that these devices will be one day operational in the human bodies. For this reason, the effective actuation of microrobots and also their positioning during the procedure are of very high importance. I fully support publication of this review. it is clearly organized, well written. I liked the AAS concept. Well done!

As an important review article, it can be strengthened further by:

In this review, only actuation of robots using an external energy source has been discussed, how about bubble propelled robots? As several In vivo application has been reported using bubble propelled (I remember about 8-9 papers for stomach) you may include bubble propelled or clearly mention that you have chosen external energy sources only.

Could you please explain the following argument more clearly? It’s not clear that it’s an advantage (not as harmful as the rest) or disadvantage (it’s harmful) How does harm? I would recommend adding some references to support this argument, please.

“The magnetic field has a high penetration depth and is little harmful to the human body, so it can be used for deep tissue imaging.”

I would recommend including a table and list advantages and disadvantages of each actuation method.

Reviewer 2 Report

This review paper is written under the system integration for autonomous actuation in vivo. 
The theme of review is excellent, but the system integration writing have many vague points in real time meaning. 
1. Especially, It is difficult to catch what is the meaning of contents from 6.1 section to 6.5 section exactly. 
There seems to be many duplications between section 6 and section 4 and 5. 
2. A real time tracking algorithm will be needed to track the microrobots in vision system. This vision algorithm is not described
concretely.
3. Section 4,5 and 6 were not written interactively. The contents of these sections must be rearranged to show more clear concepts. 

Round 2

Reviewer 2 Report

This review paper is improved by my guide properly. This paper reflect current technology in my opinion correctly in aspect of real time tracking and system construction. But if enough time to review is given, I maybe find more points to correct to improve this manuscript.